# MicroRNAs and Calcium Signaling in Heart Disease

**DOI:** 10.3390/ijms221910582

**Published:** 2021-09-30

**Authors:** Jae-Ho Park, Changwon Kho

**Affiliations:** 1Division of Metabolism and Nutrition, Korea Food Research Institute, Iksan 55365, Korea; jaehopark@kfri.re.kr; 2Division of Applied Medicine, School of Korean Medicine, Pusan National University, Yangsan 50612, Korea

**Keywords:** calcium signaling, microRNA, cardiac hypertrophy, heart failure, myocardial infarction, atrial fibrillation

## Abstract

In hearts, calcium (Ca^2+^) signaling is a crucial regulatory mechanism of muscle contraction and electrical signals that determine heart rhythm and control cell growth. Ca^2+^ signals must be tightly controlled for a healthy heart, and the impairment of Ca^2+^ handling proteins is a key hallmark of heart disease. The discovery of microRNA (miRNAs) as a new class of gene regulators has greatly expanded our understanding of the controlling module of cardiac Ca^2+^ cycling. Furthermore, many studies have explored the involvement of miRNAs in heart diseases. In this review, we aim to summarize cardiac Ca^2+^ signaling and Ca^2+^-related miRNAs in pathological conditions, including cardiac hypertrophy, heart failure, myocardial infarction, and atrial fibrillation. We also discuss the therapeutic potential of Ca^2+^-related miRNAs as a new target for the treatment of heart diseases.

## 1. Introduction

Heart disease (HD) is a major concern in global health. The number of patients with heart disease worldwide has nearly doubled in the last 10 years [1]. According to the latest update, HD remains the leading cause of death worldwide [1,2]. HD is a major economic burden. The global cost of medical care for heart failure (HF) has been estimated to increase from approximately USD 863 billion (2010) to USD 1044 billion (2030) [3,4].

The coronavirus disease 2019 (COVID-19) is spreading worldwide, with over 170 million infected and 3.7 million deaths across nearly 200 countries. People with HD or those who have had a stroke are associated with an increased risk of severe complications from COVID-19 [5]. There is growing evidence that COVID-19 is more deadly for patients with HD, including acute myocardial infarction (MI), arrhythmias, and HF [5,6]. According to experts, COVID-19 infection may directly affect heart health and mortality rates for many years owing to increased lifestyle-related risks during and after the pandemic [1]. In 2020 (during the COVID-19 pandemic), US HD deaths increased by 4.8%, the largest increase in HD deaths since 2012 [7]. To date, except for invasive and expensive surgical procedures, there is no complete treatment to prevent or cure HD progression. Understanding the molecular mechanisms underlying the pathogenesis of HD is valuable for the development of effective therapeutic strategies and novel therapeutic targets for HD.

In the heart, calcium ions (Ca^2+^) are critical for regulating muscle contraction, and abnormal Ca^2+^ homeostasis in cardiac cells is critical in the pathogenesis of common HD. In this review, we will describe the role of the main Ca^2+^ handling proteins in cardiac function and recent findings related to the mechanism of regulation of Ca^2+^ homeostasis via microRNAs (miRNAs) in diseased hearts and their potential for HD therapy.

## 2. Roles of Calcium in Heart Contractility

Ca^2+^ is an important signaling molecule in all cell types and regulates the fundamental functions of various organs. The heart uses Ca^2+^ to maintain the cardiac rhythm and muscle function. Ca^2+^-induced signals also cause cardiac cell damage or death due to hypoxia. The heart pumps blood throughout the body as a key organ in the circulatory system. The adult human heart is composed of many types of cells that combine to form complex structures. Cardiac muscle cells or cardiomyocytes occupy ~75% of the structural volume [8] and are responsible for permanent blood flow by producing contractile force in intact hearts. Approximately three billion cardiomyocytes are activated by both electrical and mechanical stimuli to contract simultaneously.

As the typical ratio of the Ca^2+^ concentration to cytosolic concentration in the external cellular environment is close to 1:20,000, tight control of Ca^2+^ access to cells and efficient means of pumping Ca^2+^ are required. Changes in its concentration are responsible for many metabolic processes in the human body. For each beat, the intracellular free Ca^2+^ concentration in cardiomyocytes ([Ca^2+^]_i_) increases above 1 µM, allowing interaction between the contractile elements during systole and diastole, and [Ca^2+^]_i_ decreases to approximately 100 nM, causing dissociation of the contractile elements [9]. It induces relaxation, allowing the heart to refill the blood. To elevate [Ca^2+^]_i_, extracellular Ca^2+^ enters the cytoplasm across the plasma membrane (sarcolemma), or intracellular Ca^2+^ is released from Ca^2+^ storage organelles, including the sarcoplasmic reticulum (SR). In cardiomyocytes, the sarcolemma Na^+^/Ca^2+^ exchanger (NCX) is the primary mechanism by which Ca^2+^ is released from cells. Activation of the sarco/endoplasmic reticulum Ca^2+^-adenosine triphosphatase 2 (SERCA2) pump promotes the reuptake of cytosolic Ca^2+^ into the SR. Numerous other molecules are involved in Ca^2+^ homeostasis in cardiomyocytes [10]. Cellular extrinsic and intrinsic signals influence cardiac function by modulating the magnitude and timing of Ca^2+^ transients in various cardiac regulating pathways. A summary of Ca^2+^ cycling observed in cardiomyocytes is presented in Figure 1.

### 2.1. Excitation–Contraction Coupling

The contraction movement of hearts is carried out by excitation–contraction coupling (ECC), which occurs in cardiomyocytes through Ca^2+^ regulation [10]. Ca^2+^ regulation in cardiomyocytes is maintained by the action of specific proteins, such as Ca^2+^ channels, pumps, transporters, and exchangers. The SR serves as an important repository and sinks with the regulation of [Ca^2+^]_i_ during ECC, and in cardiomyocytes, most Ca^2+^ is stored in the SR within the milli-molar range. Ca^2+^-buffering proteins (e.g., calsequestrin, CSQ) within the SR lumen are important Ca^2+^ sensors and contribute to deciphering Ca^2+^ transients and signals during ECC.

Cardiac ECC is the process whereby an electrical stimulation (i.e., excitation) of the surface membrane with an action potential (AP) triggers a cardiomyocyte to depolarize and contract. Ca^2+^ plays two pivotal roles in the ECC. Ca^2+^ drives myofilament activation and regulates sarcolemma ionic currents that are responsible for normal electrical rhythms. A series of ECC events in cardiomyocytes are as follows: (1) initiation and propagation of an AP along with the plasma membrane, (2) rapid spread of the potential along with the transverse tubule system (T-tubule system), (3) dihydropyridine receptors (DHPR, L-type Ca^2+^ channel CaV1.1, LTCC)-mediated detection of changes in membrane potential, (4) ryanodine receptors (RyRs) stimulation due to allosteric interaction of the LTCC with the SR RyRs, (5) release of Ca^2+^ through stimulated RyRs in SR and transient increase in [Ca^2+^]_i_, (6) transient activation of the cytosolic Ca^2+^ buffering system and the contractile apparatus, followed by (7) disappearance of Ca^2+^ from the cytosol mediated by its movement to the mitochondria via the mitochondrial Ca^2+^ uniporter (MCU), its expulsion by the NCX and plasma membrane Ca^2+^–ATPase (PMCA) at the sarcolemma, and its final reuptake by the SR through the SERCA2. Both the duration and intensity of cardiac AP affect the regulation of Ca^2+^ fluxes and contractility in cardiomyocytes.

In addition, ECC is regulated by multiple signaling pathways. The β–adrenergic pathway is regulated by β-agonists, such as adrenaline, which activate the β–adrenergic receptor (βAR) and initiates the production of cyclic adenosine monophosphate (cAMP) by adenylate cyclase, which activates protein kinase A (PKA). Another pathway is the Ca^2+^-mediated calmodulin (CaM)-dependent kinase (CaMK) signaling, which is activated by increased cytosolic Ca^2+^content, thereby inducing regulation of ECC. More recent studies have reported a process called excitation–transcription coupling (ETC), which reveals a similar role of Ca^2+^ in controlling gene transcription in cardiomyocytes to that of Ca^2+^-dependent signaling of excitation–contraction coupling [11].

### 2.2. Two Major Players in SR Calcium Flux in Heart Disease

ECC defects that cause contractile dysfunction and cardiac arrhythmia are typical features of HF. In failing cardiomyocytes, a major defect in Ca^2+^ cycling occurs at the SR. Abnormal expression and function of key Ca^2+^ handling proteins, including SERCA2a, RyR2, phospholamban (PLN), CaM, CSQ2, triadin, and junctin, result in excessive [Ca^2+^]_i_, decreased SR Ca^2+^ uptake, increased SR Ca^2+^ leak, and decreased SR Ca^2+^ content. SERCA2a and RyR2 are primarily responsible for the sequestration and release of SR Ca^2+^. Both SERCA2a and RyR2 are regulated by post-translational modifications (PTMs) and interacting partner proteins. Therefore, strategies to restore impaired SR Ca^2+^ homeostasis caused by an abnormality in these two Ca^2+^ handling proteins have been extensively studied for the treatment of HF. The major SR Ca^2+^-handling proteins reported to be associated with HD are summarized in Table 1.

#### 2.2.1. SERCA2a Calcium Pump

One of the most striking cellular changes in failing human hearts is an increase in end-diastolic [Ca^2+^]_i_ and prolongation of diastolic Ca^2+^ decay. SERCA2a is the dominant SERCA isoform in hearts and determines the clearance of more than 70% of cytosolic Ca^2+^ in humans and 90% of cytosolic Ca^2+^ in rodents [10]. Dysregulation of SERCA2a is a hallmark of HF. Several studies have shown that SERCA activity is diminished in failing human and animal hearts. Indeed, normalization of SERCA2a expression has been shown to significantly improve contractility and Ca^2+^ homeostasis in failing human cardiomyocytes [28] and increase hemodynamics with antiarrhythmic effects in rodent and large animal models of HF [29,30]. Therefore, restoration of SERCA2a function is an attractive therapeutic approach. SERCA2a gene therapy, which increases gene expression, and small molecule drugs, which stimulate enzyme activity, are being developed as new treatments for chronic HF. Several clinical studies have been conducted to correct SERCA2a enzyme abnormalities, such as CUPID (Calcium Upregulation by Percutaneous Administration of Gene Therapy in Cardiac Disease) 1, CUPID 2, AGENT-HF (AAV1-CMV-Serca2a Gene Therapy Trial in Heart Failure), and SERCA-LVAD trial [31,32,33,34]. These studies showed that SERCA2a gene delivery is safe and has potential benefits in advanced HF.

The typical underlying mechanism for reduced SERCA2a activity is inhibition by interacting with partner proteins. PLN is a reversible regulator of SERCA2a. PLN binds directly to SERCA2a and inhibits its affinity for Ca^2+^. The inhibition of SERCA2a by PLN is regulated by the phosphorylation of PLN rather than by changes in PLN expression. It has been observed that PLN phosphorylation is reduced in the heart tissues of most patients with HF [35,36]. βAR stimulation leads to PLN phosphorylation at the serine-16 site by PKA and threonine-17 site by CaMKII or protein kinase B (AKT). Phosphorylated PLN loses its binding to SERCA2a, which in turn increases SERCA2a pump activity. In addition, several PLN-binding proteins, such as hematopoietic lineage cell-specific protein-1 associated protein X-1 (HAX-1), intra-luminal histidine-rich Ca^2+^ binding protein (HRC), S100A1, and protein phosphatase 1 (PP1), contribute to PLN-dependent SERCA2a regulation [37]. In 2016, Nelson et al. discovered a muscle-specific long noncoding RNA called the dwarf open reading frame (DWORF) as a novel activator of SERCA. The DWORF peptide has been proposed to indirectly activate SERCA2a by displacing PLN [38]. However, more diverse studies are needed to determine the exact physiological role of DWORF in the heart. Some studies have reported a reduction in PLN mRNA levels in patients with dilated or ischemic cardiomyopathy [17]. Notably, pathogenic variants in PLN, known to be associated with hereditary dilated cardiomyopathy (DCM) with HF, have been reported to cause problems with binding affinity for SERCA2a. Genetic correction of PLN mutations via genome editing techniques combined with gene transfer produced positive results, including normalized Ca^2+^ handling in a patient-derived cell model of DCM, suggesting a novel strategy for DCM treatment [39]. In particular, with strong support from the Leducq Foundation, an international network of excellence program in cardiovascular research, PLN-induced cardiomyopathy studies have become intensive [40].

SERCA2a function is also regulated by PTMs, including nitrosylation, glutathionylation, glycation, SUMOylation, and acetylation. In particular, SUMO1 deficiency and decreased SUMOylation levels of SERCA2a have been observed in failing hearts [41]. Restoration of SUMO1 via gene transfer has explored the therapeutic potential of targeting SUMOylation as a method to increase SERCA2a activity and improve cardiac contractility in both mouse and porcine models of HF [41,42,43]. Furthermore, this PTM framework provides a new perspective on SERCA2a modulation, resulting in the identification of novel SUMO-SERCA2a activators [44]. Recent work has uncovered additional roles of deacetylation/acetylation in modulating SERCA2a function during HF [45]. Overall, these PTMs are considered a fine-tuning mechanism of SERCA2a, which may enhance the capacity of active targeting SERCA2a to treat HF.

#### 2.2.2. RyR2 Calcium Release Channel

RyR2 is a gate-keeping mechanism that serves as a Ca^2+^-induced Ca^2+^ release channel on the SR during the systole of cardiomyocytes. The amount of SR Ca^2+^ released by RyR2 is correlated with the strength of the systolic contraction. RyR2 is the largest ion channel in nature (four monomers of 565 kDa each). RyR2 combines with numerous accessory proteins to form a large macromolecular complex, including FKBP (FK506-binding protein 12.6), CaM, CSQ2, junctin, triadin, HRC, S100A1, and Sorcin [46]. They control the Ca^2+^ sensitivity of RyR2 through dynamic interactions, which in turn alters the probability of RyR2 opening or closing. For example, loss or dysfunction of CSQ2, the main SR Ca^2+^-binding protein, exposes RyR2 to excess free SR Ca^2+^. As another example, defective CaM binding to RyR2 destabilizes the channel. Indeed, several CaM mutations are associated with severe forms of long QT syndrome (LQTS) and catecholaminergic polymorphic ventricular tachycardia (CPVT) [22]. 

In addition, the gating ability of RyR2 is regulated by various PTMs, including phosphorylation, oxidation, and nitrosylation [47]. Many studies have shown that SR Ca^2+^ leaks are caused by increased sensitivity of RyRs to Ca^2+^ due to RyR phosphorylation by kinases (i.e., PKA or CaMKII) or phosphatases (i.e., phosphatase 1 or 2A). In particular, phosphorylation of RyR2 to specific serine residues, such as Ser-2808, Ser-2814, and Ser-2030, appears to induce functional changes in RyR2, and an association with HD has been reported [48]. For example, PKA-mediated RyR2 induces the dissociation of FKBP, a RyR2 stabilizer, which increases the probability of opening. In this context, prolonged PKA phosphorylation may lead to premature contractions and arrhythmias. However, contradictory results have been reported for RyR2-mediated Ca^2+^ spark via phosphorylation in intact cardiomyocytes [48,49].

Under pathogenic conditions such as HF, RyR2-mediated SR Ca^2+^ release continues during diastole, reducing SR Ca^2+^ content. RyR2 leakage can be due to altered expression levels of RyR2-associated proteins or dysregulation of PTMs, such as enhanced PKA- and CaMKII-dependent phosphorylation and reduced phosphatase activity. The exacerbating effects of SR Ca^2+^ leakage on cardiac function include: (1) decreased systolic SR Ca^2+^ levels leading to systolic dysfunction, (2) elevation of diastolic Ca^2+^ leading to diastolic dysfunction, (3) energy outflow to re-pump Ca^2+^, and (4) induced arrhythmias. Although there is no change in the protein expression of RyR2 in patients with HF, abnormal Ca^2+^ sparks are observed in failing cardiomyocytes [50]. The exact mechanisms and regimes of SR operation that generate abnormal Ca^2+^ leaks remain elusive. Several abnormal functions of RyR2 have been identified in patients with CPVT, arrhythmic right ventricular cardiomyopathy (ARVC), or atrial filtration (AF) as the causes of disease-related mutations and interacting partners [12,13,14,51].

Regarding therapeutics, RyR2 gene replacement approaches are limited owing to the vector size. Alternatively, siRNA delivery to silence mutant mRNA of RyR2 in an allele-specific manner or genome-editing approach has proven effective in normalizing cardiac electrophysiology in a mouse model of CPVT [52]. Several types of drugs have been developed that target mutated or dysfunctional RyR2 channels, including benzothiazepine derivatives (K201 and S107), tetracaine derivatives (EL9 and EL20), and unnatural verticilide enantiomers [53]. The potential of K201 for the treatment of AF has been evaluated clinically; however, the results have not been reported. Several FDA-approved drugs modulate RyR2, including flecainide and propafenone, which are anti-arrhythmic agents that act as Na^+^ channel blockers with additional activity in the open state of RyR2. Dantrolene, a pan-RyR inhibitor, has been investigated for its anti-arrhythmic efficacy in patients with CPVT (NCT04134845) [53].

### 2.3. Calcium Signaling through Protein Kinases

As described above, Ca^2+^ cycling is achieved through harmony with vital Ca^2+^-handling proteins strictly regulated by changes in PTMs, such as phosphorylation. The most relevant modulator for these proteins is PKA, which can directly phosphorylate and regulate major proteins in cardiomyocytes, including LTCC, RyR2, and PLN. A further mechanism of control is provided by CaMKII, targeting the same Ca^2+^ handling proteins.

### 2.3.1. β–Adrenergic Receptor-Mediated PAK Regulation

PKA is a cAMP-dependent protein kinase that has multiple roles in the regulation of cardiac function, including contraction, metabolism, ion flux, and gene transcription. A principle underlying the mechanism of cardiac β–adrenergic receptor (βAR) signal transduction is as follows: in response to stress, the binding of agonist βARs selectively interacts with the stimulatory G protein to directly stimulate adenylyl cyclase, converting ATP to cAMP, which activates PKA. βAR-mediated PKA activation phosphorylates several ECC proteins such as LTCC, RyR, cardiac troponin I (cTnI), cardiac myosin binding protein C (cMyBPC), and PLN. An additional critical target in Ca^2+^ cycling is inhibitor-1 (I-1), which is also controlled by PKA. I-1 becomes active upon PKA phosphorylation and inhibits type 1 serine/threonine protein phosphatase (PP1), resulting in amplification of βAR responses in hearts.

Chronic HF is associated with increased sympathetic nervous system activity [54]. In the early stages of reduced cardiac function, an increase in sympathetic activity preserves cardiac output. As heart function deteriorates, activation of neurohumoral signaling pathways increases to compensate for disease progression. However, prolonged neurohormonal activation causes significant damage to cardiomyocytes. Under long-term stimulation, βAR signaling is regulated by the coordinated action of at least three enzymes, both at the receptor level and downstream of the cascade: G protein-coupled receptor kinases, which phosphorylate and desensitize the receptor; cyclic nucleotide phosphodiesterases, which degrade cAMP; and phosphatases, which dephosphorylate phosphoproteins. Chronic stimulation of βAR results in multiple changes in the ARAR signaling cascade, including downregulation of βAR, upregulation of βAR kinase, and increased inhibitory G-protein α-subunit function. In this context, it is a foregone conclusion that abnormalities in the βAR-PKA pathway are important determinants of cardiac dysfunction and HF. Reduced PKA activity and decreased phosphorylation of downstream targets such as PLN, cTnI, and cMyBP, have been observed in patients with AF, hypertrophic cardiomyopathy, and HF [55,56,57]. However, increased protein levels and activity of PKA have also been reported in failing hearts [58]. PKA-mediated hyperphosphorylation of RyR2 is found in dysfunctional human and canine hearts, associated with RyR2 channel instability creating leaky channels. Although there are contradictory reports on the expression level and activity of PKA in diseased hearts, aberrant PKA activation or inactivation contributes to the pathogenesis of myocardial ischemia, hypertrophy, and HF.

### 2.3.2. Calcium-Calmodulin Mediated CaMKII Regulation

CaMKII is another critical regulatory kinase responsible for the phosphorylation of key ECC proteins and the transcriptional activation of pathological hypertrophy [59,60]. CaMKII is inactive in normal status, but is activated by increased [Ca^2+^]_i_ and reactive oxygen species (ROS). Initially, the elevation of [Ca^2+^]_i_ triggers Ca^2+^binding to CaM, activating CaMKII. Activation of CaMKII by Ca^2+^-CaM depends on the local Ca^2+^ level and the frequency of Ca^2+^ release. When [Ca^2+^]_i_ increases briefly, CaMKII returns to its inactive form after Ca^2+^-CaM dissociation. However, the continuous presence of Ca^2+^-CaM allows for autophosphorylation of CaMKII, which causes “CaM trapping” to maintain CaMKII activity even at low [Ca^2+^]_i_ conditions. For example, CaMKII can be activated through PTMs induced by ROS independent of Ca^2+^-CaM but also activated in response to βAR/PKA stimulation [61].

CaMKII is considered a major pathogenic signaling molecule in HD [62,63,64]. For example, upregulation of CaMKII activity and expression appears to be a common hallmark of cardiomyopathy of various etiologies in patients and animal models, suggesting that CaMKII is a signaling molecule in cardiomyopathy. Given the vital role of CaMKII in ion channel regulation, CaMKII seems to behave as a pro-arrhythmogenic protein in the heart. The expression and activity of CaMKII increase in AF, resulting in the promotion of arrhythmogenesis. Besides arrhythmias, oxidized, constitutively active CaMKII has been strongly linked with ischemia/reperfusion injury (I/R), diabetes cardiomyopathy, and HF.

## 2.4. MicroRNAs as a New Modulator of Calcium Signaling Pathway

MicroRNAs (miRNAs) are small, noncoding RNAs consisting of approximately 18–24 nucleotides that regulate gene expression and function at the post-transcriptional and translational levels. These molecules are expected to alter expression by 20–30% of all mammalian protein-coding genes [65]. A single miRNA can regulate multiple genes as targets, and many miRNAs can target a single gene. In addition, miRNAs can indirectly affect the expression of other miRNAs. Recent miRNA studies often (1) define expression patterns by microarray profiling or random sequencing; (2) identify downstream targets through bioinformatic analysis with in vitro validation; and (3) determine miRNA-related phenotypes.

In hearts, miRNAs broadly demonstrate their roles in physiological and pathological events [66,67]. Specific miRNAs can influence several aspects of the onset and progression of HD, such as pathological hypertrophy, fibrosis, inflammation, apoptosis, and oxidative and hypoxic damage. Cardiac Ca^2+^-handling proteins and signaling pathways are also regulated by miRNAs [68]. In addition, circulating miRNAs have also been shown to be promising biomarkers for HD [69]. Current studies suggest that the expression profile of many cardiovascular-related miRNAs may be altered by diet [70,71]. In this section, we focus on miRNAs related to Ca^2+^ homeostasis. The important miRNAs implicated in the pathophysiology of HD regulating Ca^2+^ homeostasis, along with their targets, are summarized in Table 2 and Figure 2.

### 2.4.1. Calcium Regulating MicroRNAs Related to Hypertrophy and Heart Failure

Cardiac hypertrophy (HT) is an abnormal enlargement or thickening of the heart muscle that occurs as an adaptive response to hemodynamic overload, which increases contractility and reduces ventricular wall stress. This adaptive hypertrophy transits to HF through pathological remodeling [75]. Pathological cardiac hypertrophy is commonly observed in patients with hypertension or heart valve stenosis. The major signaling pathway for pathological hypertrophy is Ca^2+^-dependent hypertrophic signaling such as angiotensin II (ET-1)/Gaq/calcineurin/NFAT [76]. The critical role of the SR Ca^2+^ transport system in HT/HF has been consistently demonstrated both in transgenic animals and by gene therapy to improve impaired Ca^2+^ handling. The relationship between pathological hypertrophy-related miRNA processes has also been extensively studied.

Early studies on Dicer knockout, a critical enzyme of miRNA biogenesis, have implicated a critical role for miRNAs in cardiac development and physiology. Da Costa Martins et al. reported that cardiac-specific deletion of Dicer leads to pathogenic cardiac remodeling and functional defects [77]. In 2006, Van Rooij et al. first reported more than a dozen miRNAs regulated in cardiomyocytes during HT or HF in mice and humans [78]. In this study, the expression of miR-24, miR-125b, miR-195, miR-199a, and miR-214 increased in both the hearts of patients with end-stage HF and in HF mice, indicating a general response pattern of adverse cardiac remodeling. To date, the regulatory function of specific miRNAs associated with cardiac HT/HF has been studied, and the list of disease-related miRNAs continues to grow. For example, miR-1, miR-9, miR-19b, miR-21, miR-23a, miR-26, miR-29, miR-98, miR-132, miR-133, miR-155, miR-195, miR-199a/b, miR-208, and miR-499 are known to contribute to pathological hypertrophy.

**miR-1** is one of the best-characterized miRNAs to date. In general, miR-1 is present in clusters with miR-133, and both miR-1 and miR-133 are among the most abundantly expressed miRNAs in hearts. miR-1 expression is linked to several transcription factors, including serum response factor (SRF), myoblast determination protein 1, and myocyte enhancer factor-2, which may be promoted by βAR activation-mediated PKA signaling [79]. To assess cardiomyocyte Ca^2+^ homeostasis, gap junction protein alpha 1 or connexin 43 (*Gja1*), potassium inwardly rectifying channel subfamily J member 2 or Kir2.1 (*Kcnj2*), sorcin (*Sri*), protein phosphatase 2 regulatory subunit B’alpha or PP2A-B56α (*Ppp2r5a*), NCX1 (*Ncx1*), and calmodulin (*calm*) were predicted and characterized as target genes of miR-1. A reduction in miR-1 expression in failing hearts has been reported in many studies. Ikeda et al. reported that miR-1 is downregulated in tissues of patients with dilated cardiomyopathy and aortic stenosis, and tends to be downregulated in ischemic cardiomyopathy hearts [80]. Cardiomyocyte-specific miR-1 overexpression was found to have anti-hypertrophic properties. Karakikes et al. showed that AAV9-mediated miR-1 gene delivery restores intracellular Ca^2+^ transient rates and kinetics and improves contractile dysfunction in rats with pressure-overload-induced HT [81]. They concluded that the beneficial effects of miR-1 are due to fine-tuned calcium metabolism through normalization of the *Ncx1/Serca2a* ratio. In contrast, miR-1 downregulation during HF suppresses miR-26 expression by increasing calcineurin–NFAT activity. The miR-26 family has been demonstrated to be an essential regulator of cardiac remodeling. The expression of miR-26 family has been observed to be downregulated in experimental animals and human patients with HF [82].

**miR-22** is abundantly expressed in hearts; however, an altered expression of miR-22 was observed in diseased hearts. miR-22 plays an essential role in regulating the transition from HT to dilated cardiomyopathy in response to pathological stresses, such as β-adrenergic stimulation or cardiac pressure overload. Gurha et al. reported that genetic deletion of miR-22 accelerates pathogenic cardiac remodeling and HF development [83]. miR-22 knockout cardiomyocytes exhibited SR Ca^2+^ handling abnormalities, particularly SERCA2a transporting activity, *Serca2a* levels, and decreased PLN phosphorylation. Several miR-22 target genes have been reported, including sirtuin 1 and purine-rich element-binding protein B (PURB) [83,84]. PURB is a repressor of SRF, a transcriptional regulator of the *Serca2a* gene, and hypertrophic genes.

Wahlquist et al. found that upregulation of **miR-25** in failing hearts inhibited the expression of SERCA2a [85]. In mice, miR-25 overexpression reduced cardiac contractile properties, but conversely, miR-25 inhibition by injection of anti-miR-25 oligonucleotides (antagomiR) improved cardiac function and animal survival in the setting of HF. Further studies showed that miR-25 could be efficiently inhibited over a long time by administering adeno-associated virus (AAV) vectors encoding an anti-miR-25 tough decoy (TuD) in vivo [86]. The protective effects of miR-25 inhibition were observed during the progression of HF. These studies suggest the potential for HF treatment by targeting miR-25, due to its ability to modulate the clinically validated target SERCA2a. In addition, Let-7, miR-24-3p, miR-133a-3p, miR-140, miR-141-3p, miR-142-3p, miR-148-3p, and miR-153-3p have been reported as potential regulators of SERCA2a in HD.

Interestingly, another miRNA involved in the regulation of the SERCA2a pump function is **miR-146a**. miR-146 has been implicated in the immune response. In immune cells, miR-146 expression is induced upon inflammatory stimuli, and it is also an NF-κB transcriptional target. Several studies have indicated the role of miR-146a in the pathogenesis of HD. Oh et al. reported the mechanical association of miR-146a with hypertension-induced cardiac HT and HF [74]. miR-146a transcript levels increased in both animals and HF patients and were inversely correlated with *Sumo1* expression, a positive regulator of the SERCA2a pump. In in vitro experiments, the decay of the Ca^2+^ transient was prolonged in miR-146a overexpressed cardiomyocytes due to decreased sumo1 expression. AAV-mediated cardiac miR-146a overexpression caused contractile dysfunction and reduced SUMOylated SERCA2a and SERCA2a protein itself, as well as SUMO levels. However, restoration of SUMO1 and SERCA2a molecules through TuD-mediated miR-146a inhibition improved cardiac function and hemodynamics and suppressed cardiac remodeling in mice subjected to transverse aorta constriction. Another study found that pharmacological inhibition of miR-146a via antagomir injection attenuated cardiac dysfunction and inhibited cardiac remodeling in rats with MI-induced HF [87]. Previous studies have shown that miR-146a can be used as a potential biomarker to evaluate the efficacy of anti-HF agents [88].

SERCA2a is also a target of **miR-328** and has been studied in in vitro and in vivo models of HT. Li et al. reported that miR-328 increased upon HT stimuli and demonstrated that the overexpression of miR-328 promoted cardiac HT accompanied with decreased SERCA2a expression and increased [Ca^2+^]_i_ [89]. In addition, miR-328 alters the LTCC current by directly targeting the LTCC subunits, CCNB2, thereby increasing the susceptibility of the heart to arrhythmia. The expression of miR-328 increased 3.5 times in atrial tissue in AF patients [90].

Expression levels of the **miR-212/miR-132** cluster were upregulated in cardiomyocytes under hypertrophic conditions. The miR-212/miR-132 family is a calcineurin/NFAT signaling cascade regulator that directly downregulates the anti-hypertrophic Forkhead box protein O3 (*Foxo3*) transcription factor. Ucar et al. reported that miR-212/132 knockout mice protect cardiac function from pressure-overload-induced HF, whereas cardiac miR-212/132 overexpressing mice displayed pathological HT and HF phenotypes [91]. Recent studies have shown that the miR-212/132 family also regulates SERCA2a by targeting the 3’-UTR of the *Serca2a* gene [92]. Preclinical studies have been conducted to develop anti-miR-132 therapeutics. Foinquinos et al. reported the therapeutic efficacy of a synthetic antisense oligonucleotide inhibitor against miR-132 (antimiR-132) in a pig model of post-MI [93]. CDR132L, the first antimiR-132 agent, recently completed clinical evaluation in a Phase 1b study (NCT04045405) that demonstrated safety and acceptable pharmacokinetics and is suggested to improve cardiac function in patients with HF [93].

Additionally, it was reported that dysfunction of miR-30, miR-148/152, and miR-625 is likely to be associated with Ca^2+^-dependent signaling during cardiac HT and HF through the inhibition of CaMKII.

### 2.4.2. Calcium Regulating MicroRNAs Related to Ischemic Heart Disease

Ischemic heart disease or myocardial infarction (MI) is the most common underlying cause of HF. In pathological hypertrophy, adverse cardiac remodeling after myocardial ischemia eventually leads to cardiac dysfunction and decreased performance. Among the complex signaling networks that characterize myocardial remodeling, the distinct processes are cardiomyocyte loss, cardiac hypertrophy, alteration of extracellular matrix homeostasis, fibrosis, autophagy defects, metabolic abnormalities, and mitochondrial dysfunction.

Although reperfusion strategies to reduce ischemic injury are widely used in clinics for MI, reperfusion following ischemia (I/R) injury is an important complication of this therapy. Cardiomyocyte death, arrhythmias, and contractile dysfunction are the main signs of myocardial I/R injury. Defects in the myocardial Ca^2+^ transport system with cytosolic Ca^2+^ overload are significant contributors to myocardial I/R injury [94]. Reductions in SERCA2a activity and SR Ca^2+^ uptake have been reported in most myocardial I/R studies. Consistently, the rate of SR Ca^2+^ reuptake significantly decreased in the human myocardium after reverse I/R. Several studies have reported that I/R-mediated cytosolic Ca^2+^ overload can be reduced by pharmacological inhibition of reverse-mode NCX activity. In addition, altered expression or activity of LTCCs and CaMKII, and oxidative modifications of Ca^2+^ handling proteins after I/R injury have been reported in experimental animals.

Many studies have suggested that miRNAs are associated with the progression of myocardial I/R injury and MI [95,96]. For example, miRNA-1, miRNA-15, miRNA-92a, miRNA-320, and miRNA-574 can have detrimental effects on I/R injured hearts. miRNA-21, miRNA-24, and miRNA-29 appear to play dual roles in the pathogenesis of cardiac I/R injury. In contrast, miRNA-126, miRNA-133, miRNA-144, miRNA-145, miRNA-199, miRNA-210, miRNA-214, miRNA-494, miRNA-451, and miRNA-499 were found to protect the heart from I/R injury.

**miR-214** is transcribed together with miR-199a-2 and is upregulated in response to various cardiac stresses, including pressure overload, MI, and excessive calcineurin, a calcium/calmodulin-sensitive phosphatase. Aurora et al. reported that miR-214 null mice exacerbated myocardial I/R injury due to loss of cardiac contractility, increased apoptosis, and excessive fibrosis [97]. Through gene expression profiling by microarrays, altered Ca^2+^ handling was confirmed in miR-214 knockout mouse hearts. miR-214 directly inhibited *Ncx1*, increasing NCX protein expression, and reverse mode activity miR-214 knockout mice. Furthermore, CaMKII, a key regulator of Ca^2+^ signaling, also contains a miR-214 binding site, and mRNA levels of CaMKII increased in miR-214 deficient hearts upon I/R compared to wildtype-controls.

**miRNA-145** is enriched in hearts and is dysregulated in animal models after I/R injury. miR-145 is downregulated in MI rats, whereas upregulation of miR-145 expression promotes the repair of infarcted myocardium. In cardiomyocytes, miR-145 inhibits ROS-induced Ca^2+^ overload and Ca^2+^-related signals by directly targeting CaMKII [98].

**miR-1** is also known to be involved in the I/R injury process [95]. While miR-1 is upregulated in remote areas of infarcted regions relative to healthy adult hearts, miR-1 was downregulated in rat cardiac tissue post-I/R. By targeting several anti-apoptotic genes, miR-1 can affect either cardiac cell death or survival depending on the pathogenic conditions. The arrhythmia promoted by miR-1 is described below.

MicroRNA dysregulation due to I/R injury involves different reactions depending on the damage process. Acute inhibition or overexpression of miRNAs after MI might be beneficial in limiting tissue damage and preventing long-term adverse remodeling and HF.

### 2.4.3. Calcium Regulating MicroRNAs Related to Atrial Fibrillation

AF is a persistent cardiac arrhythmia, especially in the elderly, which can cause or exacerbate HF. Delayed afterdepolarizations (DADs) are the most important mechanisms underlying focal atrial activity [99]. AF atria show SR Ca^2+^ handling abnormalities, triggering spontaneous diastolic SR Ca^2+^ release. However, the SR Ca^2+^ load was not increased significantly, suggesting that SR Ca^2+^ leaks occur due to altered RyR2 function. Excess diastolic Ca^2+^ is removed by NCX, creating a net depolarizing current called the transient inward current that produces DADs. Several studies have demonstrated that the level of RyR2 protein is elevated in the atria of patients with paroxysmal AF. In addition, increased NCX expression and function are commonly observed in AF. Altered Ca^2+^ signaling may also contribute to structural and electrical remodeling. For example, Ca^2+^ overload can activate the calcineurin-NFAT pathway, leading to HT and fibrosis.

The link between **miR-1** and the onset of cardiac arrhythmia has been observed in animal models and humans. miR-1 expression is significantly upregulated in ventricular arrhythmias. The possible mechanisms of hyperactivated miR-1-mediated arrhythmia include enhanced Ca^2+^ release, dissociation of phosphatase activity from the RyR2 complex, and altered expression of K^+^ channels that impairs the kinetics or membrane trafficking systems. Several studies have shown that miR-1 and miR-133 may indirectly increase calcium release by targeting B56α, a regulatory subunit of PP2A. Yang et al. found that miR-1 promotes ischemic cardiac arrhythmias by targeting the KCNJ2 gene, which encodes the Kir2.1 inward rectifier K^+^ channel protein subunit, and the *GJA1* gene, encoding connexin-43 gap junction channel protein subunit [100]. In addition, upregulation of miR-1 expression by aldosterone blocker treatment reduced the incidence of ventricular arrhythmias, in part, by targeting hyperpolarization-activated cyclic nucleotide-gated channels in MI rats [101]. In contrast, the expression of miR-1 was downregulated in patients with age-related AF and patients with advanced-stage AF undergoing cardiac surgery [102]. This increased miR-1 expression correlates with increased Kir2.1 transcript levels and inward rectifier K^+^ current density. These results suggest different effects of miR-1 under other arrhythmia conditions; therefore, the exact mechanism of miR-1 mediated arrhythmia generation requires further study.

The **miR-106b~25 cluster** consists of the highly conserved miR-106b, miR-93, and miR-25 and is downregulated in patients with paroxysmal AF, which is associated with elevated RyR2 expression. miR-106b and **miR-93** could negatively regulate RyR2-3′UTR [103]. Moreover, loss of the miR-106b-25 family induced pro-arrhythmic SR Ca^2+^ release, resulting in increased AF sensitivity in mice.

The **miR-208**
**(miR-208a/b) family** is a cardiomyocyte-specific miRNA encoded within introns of the cardiac myosin heavy chain genes (i.e., MYH6 and MYH7). Of the two isoforms, increased miR-208b was consistently observed in tissues of damaged hearts, especially MI and dilated cardiomyopathy. Canon et al. observed upregulated miR-208b expression in AF cardiomyocytes [104]. Aberrant miR-208b levels, but not miR-208a, were inversely correlated with the mRNA levels of SERCA2a predicted as a direct target gene. Overexpression of miR-208b in HL-1 atrial myocytes showed reduced SERCA2 protein levels, suggesting the involvement of miR-208b in Ca^2+^ handling impairment during atrial remodeling.

In addition, upregulated **miR-21**, **miR208b**, and **miR-328** contributed to adverse electrical remodeling in AF by targeting LTCC. In particular, miR-21 has been extensively studied for its role in atrial fibrosis during AF. Several studies have reported that circulating miR-208b and miR-328 are related to AF prevalence [105], suggesting a new biomarker for AF.

The modified miRNAs of hearts due to AF contribute to cardiac electrical and structural remodeling. A recent meta-analysis conducted by Shen et al. reported that 283 miRNAs changed compared to AF and non-AF controls [106]. Of the 51 consistent miRNAs (consistency of expression patterns), 22 miRNAs, including miR-155, miR-223, miR-483, and miR-1202, were upregulated in AF and 29, including miR-125b, miR-143, miR-145, and miR-208, were downregulated in AF.

### 2.5. miRNAs as Potential Therapeutics for Heart Disease

As our knowledge of the biosynthesis and functionality of miRNAs in hearts has increased over the past two decades, attempts have been made to identify HD-related miRNAs and understand mechanisms of action and downstream effectors for potential clinical applications. Inconsistent findings in the expression profiling of several miRNAs and miRNA-target relationships may be due to differences in HD models (types and stages of disease), sample sources, subjects of study, and technical issues. However, these discrepancies also indicate that miRNAs mediate gene regulation in a tissue-cell-specific manner for a particular disease state.

Various strategies are being used to modify miRNA levels through restoration/replacement or suppression as a potential treatment method [107]. A specific miRNA gain-of-function can be achieved by synthesizing miRNAs called miRNA mimics. miRNA mimics are double-stranded RNA oligonucleotides that include one strand identical to endogenous mature miRNA, which generally increases the efficiency of miRNA expression. In addition, plasmid- and virus-based overexpression of miRNA precursors (i.e., primary miRNAs) allows exogenous miRNA expression in specific regions and cell types. In contrast, miRNA activity can be inhibited using specific synthetic miRNA antagonists with sequences complementary to the target miRNA, such as miRNA masks, anti-miR, anti-miRNA antisense oligonucleotides (AMOs), locked-nucleic acids (LNA), or antagomiRs. A possible strategy to sufficiently target certain tissues after systemic administration is coupling the miRNA antagonists to a specific conjugate substance (e.g., GalNAc and GN3) or enveloping the miRNA antagonists in a vector (e.g., a lipid nanoparticle). Vector-based miRNA inhibitors, such as miRNA sponges, erasers, miRNA-mower, and TuD, offer alternative approaches to promote long-term expression and inhibitory efficacy. These techniques demonstrated effective inhibition of the target mRNA in vivo.

Gene therapy has often been suggested as a technique to limit local approaches rather than systemic administration, including limited vector size, inefficient delivery to target tissues, and nuclear localization requirements for protein synthesis. In this context, miR-regulated treatment based on gene therapy may be a better choice. miR regulators are much smaller, so there are no limitations on the size of the genes being delivered [108]. As miR regulators are necessary to enter only the cytosol of the target cell, whole-systemic delivery, which is also used in the siRNA approach, would be appropriate. However, there are several potential limitations in developing miRNA-based therapeutics regarding safety and efficacy, including optimizing miRNA chemical modifications and finding a suitable delivery system for a specific miRNA. Although it can be considered an advantage and a disadvantage, this is an innovative idea in terms of treating HD, a complex disease with multiple connecting signal paths. miRNAs may induce sequence-specific and nonspecific off-target effects, such as siRNAs [108], which can bind to entirely complementary nucleotide sequences and similar sequences. Meanwhile, several studies have reported that chemical modifications to siRNAs that mediate gene silencing may address these issues as they significantly reduce unintended regulation. Furthermore, miRNAs are considered less toxic than siRNAs as they are endogenous biomolecules. However, off-target studies using miRNAs are not well developed compared to siRNAs. In addition, the specificity of viral-mediated delivery issues that are not limited to miRNA therapy continues to be a challenge. Current approaches use cardiac muscle-specific promoters (e.g., cardiac troponin T or α-cardiac myosin heavy chain) and viral serotypes (e.g., AAV1 and AAV9) that have been demonstrated to be introduced into cardiac cells [109].

To date, only a few miRNA-based drugs are still in the early phases of clinical trials, but there are clinical research efforts to investigate novel miRNA drug candidates for HD along with the miR-132 study (described above) [110]. For example, miRagen is developing MRG 110, an LNA-based miR-92a inhibitor, and Remlarsen (MRG201), a synthetic miRNA-29b mimic for the treatment of HF. Two other molecules, miRagen, MGN-9103 (a cardiac-specific miR-280a inhibitor) and MGN-1374 (a miR15/195 inhibitor), are in the preclinical stage for the treatment of chronic HF. Miravirsen was the first miRNA drug to participate in human trials. Studies have shown that systemic delivery of antagomiRs targeting hepatic miR-122 is beneficial in humans without long-term safety concerns, suggesting that miRNA therapeutics remain promising. Future studies should demonstrate the safety and efficacy of optimized delivery systems focusing on the clinical application of HD-related miRNA treatments.

## 3. Conclusions

In this review, we explored critical molecules involved in Ca^2+^ homeostasis in cardiac cells and the mechanisms of Ca^2+^ regulation through miRNAs, whose functions are altered during HD. A potential benefit of miRNA-based therapies is their ability to simultaneously target many genes in a given pathway and ease of application. Given the critical role of Ca^2+^ metabolism in cardiac pathophysiology, Ca^2+^ regulating miRNAs have significant potential as therapeutic agents in HD.

## Figures and Tables

**Figure 1 ijms-22-10582-f001:**
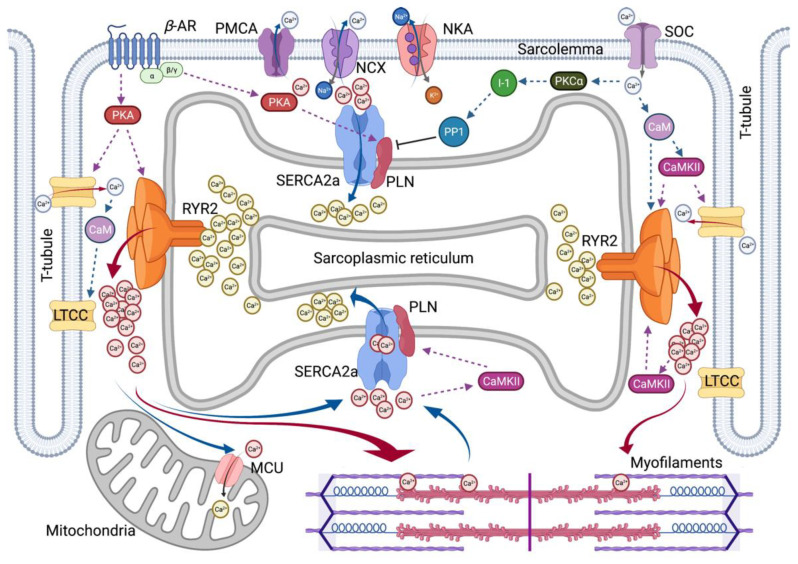
Overview of Intracellular Ca^2+^ Cycling in Cardiomyocytes. Schematic overview summarizes excitation–contraction coupling in cardiomyocytes. βAR, β–adrenergic receptor; CaM, calmodulin; CaMKII, Ca^2+^/CaM-dependent kinase II; I-1, Inhibitor-1; LTCC, L-type Ca^2+^ channel; MCU, Mitochondrial Ca^2+^ uniporter; NCX, Na^+^/Ca^2+^ exchanger; NKA, Na⁺/K⁺-ATPase; PKA, Protein kinase A; PKCα, Protein kinase C α-isoform; PLN, Phospholamban; PMCA, Plasma membrane Ca^2+^–ATPase; PP1, Protein phosphatase-1; RYR2, Ryanodine receptor type-2; SERCA2a, Sarco/endoplasmic reticulum Ca^2+^-adenosine triphosphatase 2; and SOC, store-operated channel. Red solid arrow represents Ca^2+^-induced Ca^2+^ release. Blue solid arrow represents Ca^2+^ extrusion in diastolic relaxation. Dotted lines represent a positive or negative effect on the following molecules.

**Figure 2 ijms-22-10582-f002:**
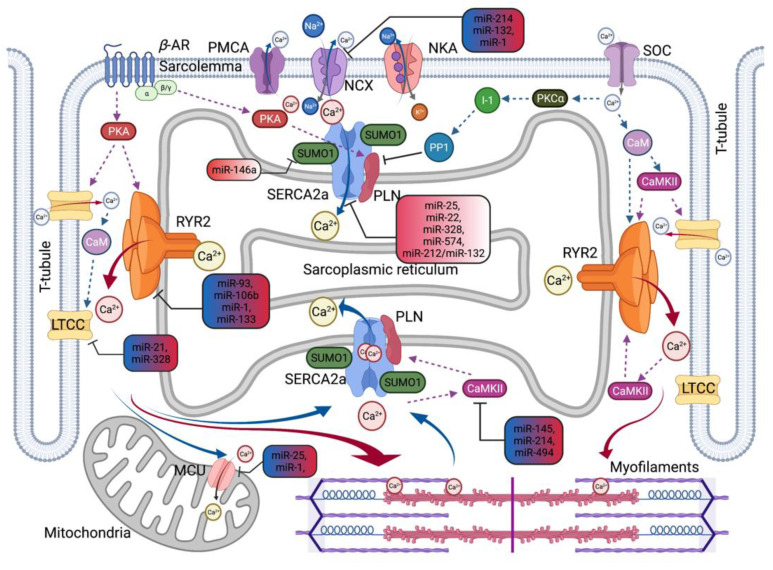
MicroRNA-Regulated Changes in Calcium Cycling in Heart Diseases. Schematic overview summarizes miR-mediated regulation of key Ca^2+^ handling proteins in heart diseases. CaMKII, Ca^2+^/CaM-dependent kinase II; LTCC, L-type Ca^2+^ channel; MCU, Mitochondrial Ca^2+^ uniporter; NCX, Na^+^/Ca^2+^ exchanger; RYR2, Ryanodine receptor type-2; SERCA2a, Sarcoplasmic reticulum Ca^2+^-adenosine triphosphatase 2; and SUMO1, Small ubiquitin-like modifier type 1. Red solid arrow represents Ca^2+^-induced Ca^2+^ release. Blue solid arrow represents Ca^2+^ extrusion in diastolic relaxation. Dotted lines represent a positive or negative effect on the following molecules. Upregulated miRNAs represent in a red gradient-filled rounded rectangle. The varying expression changed miRNAs in heart disease represent in both blue and red gradient-filled rounded rectangles.

**Table 1 ijms-22-10582-t001:** SR Calcium Handling Protein Associated with Human Heart Diseases.

Protein	Disease Phenotype	References
RyR2	CPVT, ARVD/C2, AF	Priori et al., 2001 [12]; Marks et al., 2002 [13]; Yano et al., 2005 [14]
SERCA2a	HF	Hasenfuss et al., 1994 [15]; Meyer et al., 2006 [16]; Flesch et al., 1996 [17]
PLN	ARVD, DCM	Zwaag et al., 2012 [18]; Jordan et al., 2021 [19]
CSQ2	CPVT	Lahat et al., 2001 [20]; Postma et al., 2002 [21]
CaM	CPVT, LQTS	Chazin and Johnson. 2020 [22]
Triadin	CPVT	Roux-Buisson et al., 2012 [23]; Rooryck et al., 2015 [24]
Junctin	DCM	Gergs et al. 2007 [25]
HRC	Arrhythmias, DCM, AF	Arvanitis et al., 2008 [26]; Amioka et al., 2019 [27]

AF, atrial fibrillation; ARVD, arrhythmogenic right ventricular dysplasia; ARVD/C2 arrhythogenic right ventricular cardiomyopathy type 2; CPVT, catecholaminergic polymorphic ventricular tachycardia; DCM, dilated cardiomyopathy; HF, heart failure; and LQTS, long QT syndrome.

**Table 2 ijms-22-10582-t002:** Calcium Regulating miRNAs in Heart Diseases.

microRNA ID	Pathways Related Target Genes	Ca^2+^ Homeostasis Related Target Genes	Changes in HD	References
miR-1	Cell differentiation, heart development	NCX1, RyR2, MCU	Down in HF, HT, ICM or Up in DCM	Barwari et al., 2016 [67]; Harada et al., 2014 [68]; Watson et al., 2015 [72]; Quan et al., 2018 [73]; Oh et al., 2018 [74]
miR-21	Inflammation	LTCC	Up in HT, DCM, CM	
miR-22	Apoptosis	SERCA2a	Up in HT, HF	
miR-25	Heart development	SERCA2a, MCU	Up in HF	
miR-132	Cell proliferation	NCX1	Up in HT, DCM	
miR-133	Cardiac hypertrophy	RyR2	Down in HF, HT, AF, DCM, ICM or Up in CM	
miR-145	Heart development	CaMKIIδ	Up in DCM, AS	
miR-146a	Inflammation	SUMO1	Up in HF & CHD	
miR-214	Cardiac hypertrophy	NCX1, CaMKIIδ	Up in MI, DCM, ICM, AS	
miR-328	Fibrosis	SERCA2a, LTCC	Down in HF or Up in HT	
miR-494	Apoptosis	CaMKIIδ	Down in IHD, MI	
miR-574	Apoptosis	SERCA2a	Up in MI	

AF, atrial fibrillation; AS, aorta stenosis; CHD, coronary heart disease; DCM, dilated cardiomyopathy; HF; heart failure; HT, hypertrophy; ICM, ischemic cardiomyopathy; and MI, myocardial infarction.

## Data Availability

Not applicable.

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
