# Peer review of "MicroRNAs and Calcium Signaling in Heart Disease"

_ijms, 2021, doi:10.3390/ijms221910582_

Round 1

Reviewer 1 Report

I think the authors did excellent work, proposing a very accurate update of the scientific works present, the work reads well.

Just a few points to improve:

- certainly, the part of the description of the pathophysiology is interesting and useful, but in my opinion, it should be reduced a little

- the figures are very well produced, one could be inserted on the miRNAs (general functioning, or possible targets of the analyzed miRNAs)

- one could analyze the action of diet (which has been seen to have a lot of influence on miRNAs for example doi: 10.2174 / 2211536608666181126093903) on miRNAs, especially as regards circulating ones

- also the miRNAs linked to inflammation and / to the antioxidant state are most likely modulated and will have an influence on heart diseases

Author Response

Reviewer number 1:

I think the authors did excellent work, proposing a very accurate update of the scientific works present, the work reads well.

Just a few points to improve:

  1. Certainly, the part of the description of the pathophysiology is interesting and useful, but in my opinion, it should be reduced a little.

Author response: As noted by reviewers, the pathophysiology section is an important context for this review. However, considering the length of the paper, the introduction part was shortened a bit, as suggested by Reviewer 2.

  1. The figures are very well produced, one could be inserted on the miRNAs (general functioning, or possible targets of the analyzed miRNAs).

Author response: Thank you for this suggestion. Table 2 has added columns describing the general functions of miRNAs specified in the text. Possible targets related to the calcium homeostasis of these miRNAs are presented in NEW Figure 2 as well as in table 2.

  1. One could analyze the action of diet (which has been seen to have a lot of influence on miRNAs for example doi: 10.2174 / 2211536608666181126093903) on miRNAs, especially as regards circulating ones.

Author response: Thanks for this interesting comment. A brief description was added to the text as follows: “Current studies suggest that the expression profile of many cardiovascular-related miRNAs may be altered by diet.” (See section 2.4. MicroRNAs as a New Modulator of Calcium Signaling Pathway). Two new references have been added, including the article mentioned by reviewer 1.

  1. Also the miRNAs linked to inflammation and / to the antioxidant state are most likely modulated and will have an influence on heart diseases.

Author response: This is a good point, but for the purpose of this review, it was briefly described in the text as follows: Specific miRNAs can influence several aspects of the onset and progression of HD, such as pathological hypertrophy, fibrosis, inflammation, apoptosis, and oxidative and hypoxic damage. (See section 2.4. MicroRNAs as a New Modulator of Calcium Signaling Pathway).

Reviewer 2 Report

The article by Park and Kho reviews the available literature on the impact of defects in calcium signaling and handling in heart disease. They specifically focus on the role of relatively recently discovered microRNAs and their roles in a number of pathologies and discuss the potential of Ca2+ related miRNAs as targets for heart disease treatmen.

The article covers a number of interesting issues on the level of basic science, which may be also of clinical relevance and is well written.

Points of critique:

The manuscript should be shortened. It is not needed to summarize common knowledge about heart disease in the introduction - how many patients are suffering from heart disease and what is the economic burden connected with heart problems. In addition, textbook knowledge about the role of calcium in heart contractibility can be shortened to a major degree as it has been reviewed extensively.

Specifics:

I wonder, why the authors do not mention the relatively newly discovered impact of calmodulin mutations causing heart arrhythmia and why calmodulin is not mentioned in table 1 and also not present in Fig.1 as mutations may cause CPVT and LQTS and other AF. This should be added.

Author Response

Reviewer number 2:

The article by Park and Kho reviews the available literature on the impact of defects in calcium signaling and handling in heart disease. They specifically focus on the role of relatively recently discovered microRNAs and their roles in a number of pathologies and discuss the potential of Ca2+ related miRNAs as targets for heart disease treatment. The article covers a number of interesting issues on the level of basic science, which may be also of clinical relevance and is well written. 

Points of critique:

The manuscript should be shortened. It is not needed to summarize common knowledge about heart disease in the introduction - how many patients are suffering from heart disease and what is the economic burden connected with heart problems. In addition, textbook knowledge about the role of calcium in heart contractibility can be shortened to a major degree as it has been reviewed extensively.

Author response: It was revised as requested. See introduction section.

Specifics:

I wonder, why the authors do not mention the relatively newly discovered impact of calmodulin mutations causing heart arrhythmia and why calmodulin is not mentioned in table 1 and also not present in Fig.1 as mutations may cause CPVT and LQTS, and other AF. This should be added.

Author response: Thank you for pointing it out. The involvement of the calmodulin mutations in cardiac rhythm disorders, particularly catecholaminergic polymorphic ventricular tachycardia and long QT syndrome, was described in the text (see section 2.2.2. RyR2 Calcium Release Channel), Table 1, and Figure 1. Therefore, one new related reference was added.